# Evaluation of Genome-Scale Model Reconstruction Strategies for *Lentilactobacillus kefiri* DH5 and Deciphering Its Metabolic Network

**DOI:** 10.3390/metabo15120767

**Published:** 2025-11-26

**Authors:** Maryam. A. Esembaeva, Mikhail A. Kulyashov, Tatiana S. Sokolova, Ilya R. Akberdin, Alexey E. Sazonov

**Affiliations:** Department of Computational Biology, Scientific Center of Genetics and Life Sciences, Sirius University of Science and Technology, 354340 Sirius, Russia

**Keywords:** GSM, metabolic modeling, FBA, LAB, *Lentilactobacillus kefiri*, D-gluconate

## Abstract

Background/Objectives: Genome-scale metabolic models (GSM) are key tools for predicting microbial physiology, yet species within the genus *Lentilactobacillus* remain largely unexplored. *Lentilactobacillus kefiri* DH5 is an obligately heterofermentative lactic acid bacterium with unique redox metabolism, but no curated GSM model exists for this species. This study aimed to generate the first GSM model for *L. kefiri* DH5, evaluate multiple reconstruction tools, and characterize metabolic features underlying its heterofermentative metabolism. Methods: Draft GSM models were generated from the *L. kefiri* DH5 genome annotation using five reconstruction tools. For each tool, gap-filling was performed on a CDM, followed by quality assessment using the MEMOTE. Manual curation was performed using the COBRApy library. Results: Among the five reconstructions, the KBase-derived draft demonstrated the highest quality and production potential for metabolites characteristic of heterofermentative fermentation. During manual curation of this model, reaction directions in central carbon metabolism and amino acid pathways were corrected. Analysis further identified an alternative NADH-regenerating glucose shunt via D-gluconate, supported by omics data and enzyme promiscuity considerations. Incorporation of this pathway resolved the redox imbalance and allowed the model to reproduce metabolic exchange profiles characteristic of obligate heterofermenters. Conclusions: We developed the first manually curated genome-scale model of *L. kefiri* DH5 and showed that the choice of reconstruction tool substantially affects model quality and predictive power. We also proposed an alternative glucose assimilation shunt via gluconolactone, which resolved the redox imbalance in the model and enabled representation of the heterofermentative metabolism.

## 1. Introduction

*Lentilactobacillus kefiri* is a heterofermentative lactic acid bacterium (LAB) which is most frequently found in kefir grains and fermented food, where it plays a key role in shaping the sensory profile and nutritional value of these traditional products. As a natural kefir inhabitant, *L. kefiri* demonstrates a robust fermentative metabolism, antagonistic activity against pathogens, and notable probiotic features such as immunomodulation and lactose degradation [1,2,3,4]. A growing number of data support its antimicrobial efficacy against diverse pathogenic microorganisms, including bacteria and fungi, as well as its beneficial effects in modulating host immune responses, reducing cholesterol level, and influencing metabolic functions associated with anti-obesity effects [1,2,3,4]. Although *Lentilactobacillus kefiri* is frequently reported as a promising probiotic species, the metabolic basis of its beneficial properties remains poorly understood. Existing studies mainly describe strain-specific probiotic effects but do not elucidate the metabolic mechanisms which underlie these differences between strains. Moreover, the amount of experimental data characterizing the physiology, functional properties, and internal mechanisms of *L. kefiri* is very limited, which restricts our ability to link observed phenotypes to specific metabolic pathways [5]. This knowledge gap becomes even more pronounced when focusing on the reference strain *L. kefiri* DH5: a PubMed search enabled us to retrieve only four studies [6,7,8,9]. Thus, the physiology and molecular mechanisms driving the beneficial effects of *L. kefiri* DH5 remain largely unexplored. Given the limited experimental characterization of *L. kefiri*, computational approaches become essential for understanding and deciphering its metabolism.

Genome-scale metabolic (GSM) models provide in silico representations of microbial metabolism by integrating genomic, biochemical, and physiological data. They allow simulations of metabolic fluxes under different environmental or genetic conditions, providing predictions of nutrient requirements, exploration of phenotypes, and rational design of metabolic engineering strategies [10,11,12]. Several curated GSM models, such as *i*BT721 for *Lactobacillus plantarum* WCFS1 [13], *i*NF517 for *Lactococcus lactis* MG1363 [14], and *i*LM.c559 for *Leuconostoc mesenteroides* subsp. *cremoris* [15], have enabled the identification of key metabolic features, including amino acid auxotrophies, flavor-forming pathways, and energy maintenance requirements. Recent large-scale reconstruction by Ardalani et al. (2024) [16] generated 2446 curated GSM models covering 26 *Lactobacillaceae* species, revealing species-specific metabolic traits, niche-enriched reactions, and differential auxotrophy profiles across the family. However, to date, no manually curated GSM model has been developed for *Lentilactobacillus kefiri*, limiting our ability to systematically characterize its metabolism and explore the mechanisms underlying its functional and probiotic properties. Given the metabolic versatility, functional significance, and probiotic potential of *L. kefiri*, bridging this knowledge gap through comprehensive metabolic reconstruction is a critical step forward.

Herein, we present the first manually curated genome-scale metabolic model for *Lentilactobacillus kefiri* DH5, providing a robust computational platform to investigate metabolic pathways, predict strain behavior under diverse conditions, and support targeted development for probiotic and biotechnological applications.

## 2. Materials and Methods

### 2.1. Reconstruction of the Genome-Scale Metabolic Model

#### 2.1.1. Reconstruction Procedure

The genome-scale metabolic model was reconstructed based on the published genome assembly of *L. kefiri* DH5 (ASM973953v1) from the NCBI database. Five alternative open-source tools were used for the GSM model reconstruction: KBase (www.kbase.com (accessed on 17 November 2025)) [17], Reconstructor (https://github.com/emmamglass/reconstructor (accessed on 17 November 2025)) [18], CarveMe (https://github.com/cdanielmachado/carveme (accessed on 17 November 2025)) [19], Bactabolize (https://github.com/kelwyres/Bactabolize (accessed on 17 November 2025)) [20], and PanGEM (https://github.com/omidard/LactoPanGEM (accessed on 17 November 2025)) [16]. This approach enabled the generation of several draft metabolic models based on different automatic reconstruction algorithms: reconstruction based on the annotated genome using KBase, Reconstructor, and CarveMe, which rely on different underlying databases (ModelSEED for KBase, KEGG for Reconstructor, and BiGG Models for CarveMe), and a pangenome-based models in the case of Bactabolize and PanGEM. Importantly, PanGEM was originally validated on representatives of *Lactobacillaceae*, making it particularly relevant for *L. kefiri*, and Bactabolize was included to allow a comparative evaluation of pangenome-based reconstruction approaches. We used a published pan-genome, pan-proteome, and pan-model of the *Lactobacillaceae* family, developed as part of the PanGEM project [16], which served as the basis for generating the strain-specific model of *L. kefiri* DH5. For Reconstructor, Bactabolize and PanGEM GLPK solver were used. To reconstruct the model via CarveMe, we used SCIP solver as described in the manual.

Reconstructions were performed using default parameters for Kbase (Plugin Build metabolic model version 2.2.1 with automatic template selection); for Reconstructor (version 1.1.1), CarveMe (version 1.0.5) and Bactabolize (version 1.0.5) as described in the manual, but Gram-positive template for reconstruction was employed using the -u grampos flag for CarveMe-based reconstruction and the -Gram-positive flag for Reconstructor, while the -media_type parameter was added based on CDM composition (Appendix A) for Bactabolize reconstruction. PanGEM-based reconstruction was performed using the GEMgenerator.py function using default parameters.

#### 2.1.2. Gap-Filling Procedure

After draft model generation, gap-filling was performed within each of the reconstruction tools, excluding Bactabolize and PanGEM. The gap-filling procedure was carried out under conditions simulating growth on chemically defined medium (CDM) described by Otto et al. (1983) [21] and Poolman and Konings (1988) [22], which is used for cultivation and harnessed in published GSM models for LAB [23], under anaerobic conditions. The procedure allowed the elimination of pathway gaps in the metabolic network. Detailed constraints of the reactions for CDM are presented in the Appendix A based on [15].

Gap-filling procedure in KBase, Reconstructor, and CarveMe was performed using default parameters. In the case of Reconstructor, the -type 3 flag was used, while for CarveMe, gap-filling was performed under the reconstruction process to take into account the reaction’s scores for gap-filling, as recommended in the manual. Bactabolize and PanGEM: Bactabolize gap-filling was also running with reconstruction, but the algorithm could not perform it eventually. There was no detailed pipeline description in the PanGEM guidelines in regard to gap-filling and the presented code needs modifications to be run. The CarveMe-based algorithm returns the error of gap-filling.

#### 2.1.3. Quality Assessment

All reconstructed models were exported into SBML format. Quality assessment of the reconstructed draft models was performed using the open-source software MEMOTE (version 0.17.0, Novo Nordisk Foundation Center for Biosustainability, Lyngby, Denmark) test suite [24], which is used as a standard for analysis of GSM model quality and contains a set of consensus tests.

#### 2.1.4. Benchmarking of Metabolite Exchange Profiles of Reconstructed Models

There are no measured experimental data for consumption or production rates of metabolites for *L. kefiri*. Given that, we decided to make a comparison of metabolite exchange profiles with another heterofermentative LAB—*L. mesentroides* [15]. Based on it, reconstructed models were constrained using CDM file (Appendix A) with the maximum glucose uptake rate 25 mmol·gDCW^−1^·h^−1^ and maximum uptake rate of amino acids 0.2 mmol·gDCW^−1^·h^−1^ for anaerobic conditions. Unfortunately, there was no experimental data for the disaccharide uptake rate, while the maximum uptake rate of the lactose was 13 mmol·gDCW^−1^·h^−1^ [15]. It can be assumed that the consumption rate of the disaccharides was lower than that of the monosaccharides. Exchange profiles were analyzed using flux variability analysis (FVA) implemented in the cobrapy library with GLPK solver. The substrate uptake rate for anaerobic conditions was increased in increments of 1 up to the maximum consumption rate described above. For aerobic conditions the glucose uptake was fixed as 10 mmol·gDCW^−1^·h^−1^ and the oxygen uptake rate was changed from 0 to 9 mmol·gDCW^−1^·h^−1^, increasing in increments of 1 up to the maximum consumption rate (based on [15]). To check the availability of lactate and ethanol production, the demand reactions were added into the Kbase-driven model due to the absence of transport and exchange reactions in the reconstructed original version. Then resulting fluxes in the FVA and metabolite exchange profiles are presented in Appendix A.

#### 2.1.5. GPR Comparison

To compare gene content between the reconstructed GSM models, a Venn diagram was created using the venn Python package (https://github.com/tctianchi/pyvenn (accessed on 15 September 2025)). To conduct the analysis, we took only genes which related to *Lentilactobacillus kefiri* DH5, while genes added at the gap-filling process were excluded from the analysis. Models reconstructed using Bactabolize and Reconstructor employed outdated gene identifiers from the GenBank database [25], whereas the model generated with CarveMe contained protein identifiers instead of gene identifiers. To bring all models to a unified format, a genome annotation in GFF file from RefSeq was used (Assembly GCF_009739535.1). A mapping was performed between the old_locus_tag and locus_tag fields for the Bactabolize and Reconstructor models; if a match was found, the corresponding locus_tag was included in the final gene set. In the case of the CarveMe model, a similar mapping was carried out between protein_id and locus_tag. As a result, gene sets with unified RefSeq identifiers were compiled for each model, enabling accurate gene comparison across mentioned reconstructions. The resulting code is available on BioUML via the following link: https://shorturl.at/34ke5 (accessed on 17 November 2025).

The final reconstructed models and MEMOTE reports for them are available at the BioUML platform and at the Gitlab project through the following links: https://shorturl.at/4Hi8U (accessed on 17 November 2025) and https://shorturl.at/fzuRE (accessed on 17 November 2025).

### 2.2. Manual Curation of the KBase Model

#### 2.2.1. Extension of the Reaction and Metabolite Annotation

The KBase-generated model was selected based on the analysis of the models’ MEMOTE report quality. Gene, reaction, and metabolite annotations in the selected model were extended using the CobraMod library [26]. As a result, the total MEMOTE score was improved by means of metabolite and reaction identifiers from the BiGG Models [27] database were also added to the model, which facilitates further manipulation of the model. However, the initial draft model exhibited unrealistically high growth rates and did not produce lactate, which necessitated manual curation using the COBRApy library [28].

#### 2.2.2. Growth Medium Constraints

The updated model included constraints on medium component uptakes based on the composition of the CDM which was previously used for the reconstruction of the model for *Leuconostoc mesenteroides* subsp. *cremoris* [15], with glucose supplied at 25 mmol·gDCW^−1^·h^−1^ and amino acids at 0.2 mmol·gDCW^−1^·h^−1^ consumption rates. In addition, the medium included micro- and macronutrients, as well as precursors required for cofactor biosynthesis (e.g., nicotinate ribonucleotide, riboflavin).

#### 2.2.3. Manual Curation of the Reaction and Metabolites

The reconstructed model contained 42 reactions unbalanced by mass and charge, most of which were related to fatty acid metabolism. Correction of metabolite charges and formulas based on the BiGG Models [27] database led to the number of unbalanced reactions being reduced to 4.

At the next stage, the model was tested using the MACAW tool [29] to identify reactions forming thermodynamically infeasible loops. The identified thermodynamically infeasible loops were resolved by redirecting or blocking the corresponding reactions based on the L. kefiri DH5 genome annotation and reaction direction information from the KEGG [30] and BiGG Models databases. The MACAW test and its results for the initial and final models were generated using Jupyter notebooks in the BioUML platform [31] available at the following link: https://shorturl.at/31EDr (accessed on 17 November 2025). The list of all modified reactions is presented in Appendix A.

Duplicate metabolite entries for S-acetoin, S,S-2,3-butanediol, cobinamide, galacturonate, R-lipoic acid, and R-allantoin were identified and removed from the model, along with their corresponding duplicate reactions. Duplicate entries for L-cystathionine were also detected, but not all associated reactions were duplicated. The metabolites in this case were merged to retain the unique reaction.

At the next stage, the model was expanded with reactions required for an accurate description of metabolism. To enable growth in CDM, exchange and transport reactions for ferrous iron were added. The inclusion of these reactions was justified by their presence in other LAB models. Ferrous iron is part of the biomass equation and, in the initial model, was formed from protoheme in the FCLT_2 reaction, which should proceed in the reverse direction according to the KEGG [30] and ENZYME databases [32]. Therefore, lower bounds of 0 and 1000, respectively, were set for this reaction. In addition, the ATP maintenance requirement (ATPM) reaction, which accounts for ATP expenditures not related to growth, was lacking in the initial model and was added. Transport and exchange reactions for lactate and ethanol, which were also missed in the initial reconstructed model, were introduced.

The malolactic enzyme (MALLAC) reaction, which is specific for heterofermentative LAB [33], was incorporated into the model. The gene (*DNL43_RS04425*) encoding malolactic enzyme was incorrectly associated with the malic enzyme (MEx) reaction in the original model. The acetolactate synthase (ACLS) reaction was also added, as it is required for the butanediol synthesis pathway, the production of which, along with some other flavor metabolites, was shown for heterofermentative LABs [15]. Despite the presence of a gene (*DNL43_RS05210*), this reaction was not included in the original reconstructed model, resulting in acetolactate being a dead-end metabolite. The glucose 6-phosphate isomerase (G6PI) reaction was also added to enable the formation of beta-D-glucose 6-phosphate directly from glucose 6-phosphate; the possibility of this reaction is supported by the annotation of the gene *DNL43_RS04170*.

The reaction directions in key metabolic pathways, including glycolysis and the TCA cycle, as well as nicotinamide and amino acid metabolism, were manually corrected based on the KEGG [30] and BiGG Models [27] databases. Reactions not associated with any gene introduced during the gap-filling process and exhibiting zero flux were blocked by setting both lower and upper bounds to zero. The modified reactions are provided in Appendix A.

The Escher tool [34] was used to reconstruct a metabolic map of *L. kefiri* DH5. All reconstruction and analysis steps were carried out in a Jupyter notebook using the BioUML platform [31] and are available via a web version of the platform: https://shorturl.at/4Hi8U (accessed on 17 November 2025).

#### 2.2.4. Biomass Equation Modification

Subsequently, based on a comparison between the composition of the biomass equation in our model and experimentally validated biomass equations from published models of LAB [13,15,35], redundant metabolites were removed from our biomass equation (Appendix A). In addition, metabolites consumed for the synthesis of DNA, RNA, and proteins were extracted from the overall biomass equation and assigned to separate reactions, with stoichiometric coefficients preserved as specified in the original biomass equation. It should be noted that, due to the absence of experimental data on energy costs associated with the synthesis of DNA, RNA, and proteins, ATP was not included in these individual synthesis reactions. All energy requirements for biosynthesis are instead accounted for in the overall growth equation, inclusive of demands for DNA, RNA, and protein production.

### 2.3. Analysis of Transcriptomics Data for L. kefiri

The raw reads from GSE229515 [36] were mapped to the corresponding reference genome, ASM973953v1 for *L. kefiri* DH5, using the Bowtie2 algorithm [37]. Illumina standard adapters were removed using fastp [38] (version 1.0.1, HaploX Biotechnology, Shenzhen, China) free software before mapping when necessary. The mapped reads were quantified with featureCounts [39] using gene features from the RefSeq annotation GCF_009739535.1. Reads that aligned to ribosomal genes were filtered out. Raw gene expression counts were normalized using the Transcripts Per Million (TPM) method to facilitate comparison across samples. The heatmap was constructed using the open-source seaborn library version 0.13.2 (Python 3.12).

### 2.4. Analysis for NADH/NADPH Imbalance

To further evaluate the hypothesis that the imbalance between NAD and NADP amounts significantly affects the model behavior, we introduced into the model the NAD transhydrogenase (NADTRHD) reaction, which balances the NAD/NADP ratio. However, since the genome of *L. kefiri* DH5 does not encode an enzyme capable of catalyzing this reaction, we deleted the reaction and proposed an alternative metabolic pathway for glucose catabolism that enables additional NAD generation via D-glucono-1,5-lactone. Therefore, the D-glucono-1,5-lactone lactonohydrolase (GL15LH) reaction was added to the model. The NADTRHD reaction was deleted from the model.

### 2.5. Sensitivity Analysis of the Biomass Equation

A sensitivity analysis was performed for the modified version of the model with the GL15LH reaction and modified biomass equation. The analysis was performed by means of the sequential increase and decrease in stoichiometric coefficients of the biomass components by 50% relative to their original values. It enabled us to evaluate the effect of each coefficient’s change on the predicted growth rate. When the ATP coefficient was changed, the coefficients of water, ADP, and phosphate were adjusted accordingly, in accordance with the mass balance of the ATP hydrolysis reaction (ATP + H_2_O → ADP + Pᵢ). The same approach was applied to peptidoglycan and teichoic acid components to preserve the stoichiometric consistency of their interdependent biosynthesis reactions (Appendix A).

### 2.6. Comparative Analysis of the Impact of Using Biomass Composition from Manual Curated LAB Models on Predictions of the Model

To evaluate the impact of biomass composition on model predictions, the final *i*EM644 model for *L. kefiri* DH5 was simulated using experimentally validated biomass equations from published LAB models: *L. mesenteroides* subsp. *cremoris* [15], *L. plantarum* WCFS1 [13], and *L. reuteri* JCM 1112 [35]. To implement each variant, reactions for DNA, RNA, and protein synthesis were updated to match those specified in the respective source models. Within the biomass equation itself, only the stoichiometric coefficients for ATP, ADP, H_2_O, H^+^, phosphate, DNA, RNA, and protein were adjusted accordingly. All simulations were performed for CDM with a fixed glucose uptake rate of 25 mmol·gDCW^−1^·h^−1^. Model behavior was assessed by comparing predicted growth rates and exchange profiles of key metabolite characteristic of heterofermentative LAB: lactate, ethanol, and carbon dioxide. All simulations were conducted using pFBA, and the simulation results are provided at the following link: https://shorturl.at/OoS9w (accessed on 17 November 2025).

Subsequently, a biomass equation sensitivity analysis was performed for all model variants (Appendix A) (see Section 2.5).

### 2.7. Simulation of L. kefiri DH5 Growth Under Different Substrate and Aerobic Conditions

To analyze the exchange profiles of key metabolites of heterofermentative LAB (lactate, ethanol, acetate, and carbon dioxide), we performed flux variability analysis (FVA). The glucose uptake rate was varied from 0 to 25 mmol·gDW^−1^·h^−1^ in steps of 1. Furthermore, we investigated the impact of culture growth on galactose and lactose on the metabolite exchange profiles. Lactose is the main carbohydrate in milk, and therefore its metabolism plays a crucial role in the growth of lactic acid bacteria in this environment. Galactose, formed during the breakdown of lactose, was also selected as a substrate, as *L. kefiri* demonstrates growth on both carbon sources. To reach the aim, the lower bound for the galactose uptake was varied from 0 to 25 mmol·gDW^−1^·h^−1^ (steps of 1), while the lactose uptake was considered from 0 to 13 mmol·gDW^−1^·h^−1^ (in steps of 1). Additionally, the exchange profile of the model was assessed under aerobic conditions with a fixed glucose uptake rate of 10 mmol·gDW^−1^·h^−1^ and oxygen uptake bounds varying from 0 to 10 mmol·gDW^−1^·h^−1^ in steps of 1.

## 3. Results

### 3.1. Reconstruction of Genome-Scale Metabolic Models Using Diverse Tools and Selection of the Model for Further Analysis

GSM models reconstructed using Bactabolize [20] and PanGEM [16] demonstrated the lowest MEMOTE total score [24] and did not predict growth on the CDM even after gap-filling. Therefore, they were excluded from further analysis. Although the model generated with Reconstructor [18] showed acceptable results for the total score, it was also not selected due to the inclusion of reactions through the gap-filling process based on homology principle in comparison with microorganisms distantly related to LAB. Additionally, we found out that Reconstructor did not include the charges for metabolites during the reconstruction procedure. However, manual curation and fixing of the issue led to a drop in the model quality. The CarveMe [19] model obtained the highest overall total score but was not used for further investigation. This was due to several specific shortcomings: a large number of reactions with charge imbalance (343 reactions), unbounded fluxes in the default medium (167 reactions), and 532 reactions without associated genes, which were added by gap-filling. The high total MEMOTE score was mainly attributed to more extensive annotation of reactions and metabolites, as well as the inclusion of Systems Biology Ontology (SBO) terms. At the same time, the model’s important parameters such as model consistency, mass balance, and charge balance are described in the Sub Total score, and the model with the highest value was reconstructed by KBase [17] (Table 1).

In addition, we performed a comparison of genes represented in the reconstructed models using a Venn diagram (Figure 1) (details in Material and Methods Section 2.1.5). It is important to note that the comparison was performed based on RefSeq identifiers. This required standardizing the gene identifiers in the models generated by CarveMe, Bactabolize, and Reconstructor to a common format, which led to a reduction in the number of genes used for comparison. The Venn diagram shows that Bactabolize (1 gene) and Reconstructor (10 genes) have lower numbers of unique genes, and most of the genes in the model generated by these tools are included by other reconstruction approaches. It is worth noting that the model reconstructed by Reconstructor contained a larger number of genes compared to other reconstructions (Table 1). However, only about 54% of them (402) belonged to *L. kefiri* DH5, which complicated its further use (Figure 1). The total number of genes represented in all tool-derived models was 131 genes (~20% of median number of genes in models). The largest number of unique genes was, in the KBase model, 17.9% of the total genes in the model, and in the CarveMe model, 19.7%. A lower number of unique genes were in the PanGEM version of the model. It can be related to different algorithms of gene search implemented in the corresponding reconstruction pipeline.

The three best models (KBase-, CarveMe-, and Reconstructor-generated) with the highest total scores according to the MEMOTE quality check were harnessed for FVA of the metabolic capabilities to find out which of them demonstrated the metabolic exchange profiles typical of heterophementive LABs (see details in Section 2.7). We checked the feasibility of three models to predict the growth on different substrates, such as glucose, galactose, and lactose, and growth under switching from anaerobic to aerobic conditions. Despite the gap-filling step, the Reconstructor-based model did not predict any growth on the CDM using the pFBA or FBA algorithms. Therefore, the final comparison was performed between the KBase- and CarveMe-generated models. Both reconstructions showed growth on all described substrates and under aerobic conditions (Appendix A), while only the KBase model demonstrated metabolite exchange profiles that were specific to heterophermentive LABs, indicating production of ethanol, lactate, and CO_2_ under anaerobic conditions, in contrast to CarveMe, which predicted the excretion of only CO_2_ and ethanol (Figure 2). Furthermore, the Kbase-driven model predicted a decrease in ethanol production and increase in acetate production with stable lactate production after a switch to aerobic growth conditions, which were described for another heterofermentive LAB—*L. mesenteroides* [15] (Appendix A). We additionally assessed the behavior of the CarveMe model under growth conditions on lactose and galactose, as well as under aerobic conditions. In none of these scenarios did the model show ability for lactate production, which contradicts the known metabolic characteristics of *L. kefiri*. We checked that the metabolic pathway for lactate production and lactate dehydrogenase (LDH), which is a key reaction for conversion of pyruvate to lactate, was presented in the model and was not blocked. Moreover, the CarveMe model contained the highest number of reactions without GPR associations, which could further hinder its usability in downstream analysis. Furthermore, the CarveMe reconstruction demonstrated very limited growth on lactose (ranging only from 0 to 5 mmol/gDCW/h) when used as the sole carbon source, despite the fact that lactose is the primary sugar in milk and *L. kefiri* can efficiently metabolize it (Appendix A). The model also showed a marked reduction in growth rate under aerobic conditions, which is inconsistent with the facultatively aerobic nature of lactic acid bacteria (Appendix A). Thus, the KBase-generated model was used for further elaborations.

### 3.2. Curation of the Selected GSM Model

To enhance the total score of the selected model, we enriched annotations for reactions and metabolites using the CobraMod library [26]. Although this did not alter the total score, the use of identifiers from the BiGG Models database significantly simplified subsequent model curation. Further improvements were aimed at increasing the Sub Total score. For this purpose, we removed duplicate metabolites and corrected the mass and charge balances of reactions.

The model contained duplicates for several metabolites, including S-acetoin, S,S-2,3-butanediol, cobinamide, galacturonate, R-lipoic acid, and R-allantoin, along with their associated synthesis and dissipation reactions. This redundancy led to incorrect flux distributions and affected simulation results. Consequently, the duplicate metabolites and their associated reactions were removed from the model. The duplicate for L-cystathionine had one unique reaction (CYSTGLr). To preserve this reaction, the duplicate was merged with the primary metabolite to maintain the completeness of the metabolic network.

Additionally, the original model contained 42 mass-unbalanced and 6 charge-unbalanced reactions, mostly associated with fatty acid biosynthesis. As a result of the corrections, the total number of unbalanced reactions was reduced to 4. These modifications enabled an increase in the model’s Sub Total score from 95% to 98%.

In addition to the MEMOTE assessment, the model was evaluated using the MACAW tool. The MACAW report identified reactions capable of forming thermodynamically infeasible loops. Such loops can lead to metabolite leakage from central metabolism and result in excessive generation of ATP, NAD, and NADP molecules. The presence of these loops compromises the predictive accuracy of the model by enabling incorrect flux distributions [40]. To resolve this issue, the directionality of the reactions involved in loop formation was constrained based on the following criteria: the directionality of corresponding reactions in the KEGG and BiGG Models databases, and the presence of genes encoding the associated enzymes in the genome annotation of *L. kefiri* DH5. The list of reactions forming thermodynamically infeasible loops, along with their adjusted upper and lower bounds in the model, is provided in Appendix A. The complete MACAW report is available via the following link: https://shorturl.at/31EDr (accessed on 17 November 2025).

Following critical modifications to the model, we applied constraints on metabolite uptake according to the composition of the CDM used for LAB ([21,22]; Appendix A). However, we encountered an issue: the model lacked an extracellular metabolite for ferrous iron (Fe^2+^), as well as the corresponding exchange and transport reactions, despite Fe^2+^ being a component of the biomass equation.

In the model, Fe^2+^ was being generated from protoheme in the ferrochelatase reaction (FCLT_2), which had been incorrectly defined as reversible. Data from the KEGG and ENZYME databases indicate that this reaction should proceed in the direction of protoheme synthesis from protoporphyrin and Fe^2+^. After correcting the reaction direction (setting flux bounds to 0 and 1000), the model predicted a growth rate of zero. Given that published models for LAB include dedicated exchange and transport reactions for Fe^2+^, we concluded that its uptake from the medium was biologically essential. Therefore, the corresponding reactions were added to the model. This decision was further supported by the presence of the gene *DNL43_RS00520* in the *L. kefiri* DH5 genome annotation, which encodes an iron ABC transporter permease.

However, the predicted growth rate of the updated model was 0.035 h^−1^, and no lactate or ethanol production was observed. To evaluate the capacity for production of these metabolites, demand reactions representing metabolite flux were added to the model as described above. It led to the model’s growth rate increasing to 0.408 h^−1^, and ethanol production rate was predicted as 23.53 mmol·gDCW^−1^·h^−1^, while the model still did not predict lactate excretion, but the FVA showed that it was possible (Figure 2). This version of the model will hereafter be referred to as the base model. While these demand reactions successfully restored model functionality, they represent a non-physiological mechanism for metabolite transport. For the final model, proper exchange and transport reactions for lactate and ethanol were implemented. This correction was biologically justified since *L. kefiri* DH5 is a heterofermentative LAB capable of producing both metabolites [15,41,42].

We subsequently expanded our curation to include key intracellular metabolic reactions. First, we incorporated the ATP maintenance requirement (ATPM), representing non-growth-associated maintenance (NGAM), as this essential energy drain significantly influences model behavior.

Further analysis revealed several missing core metabolic reactions. The malolactic enzyme reaction (MALLAC), characteristic of heterofermentative LAB [33], was absent despite the presence of its putative encoding gene *DNL43_RS04425*. This gene was incorrectly associated with the malic enzyme reaction (MEx), which was consequently blocked (bounds set to 0) when MALLAC was added. The model also lacked acetolactate synthase (ACLS), essential for butanediol synthesis, despite the presence of the gene *DNL43_RS05290*. Adding ACLS resolved a dead-end metabolite by incorporating acetolactate into the central metabolism. Finally, glucose-6-phosphate isomerase (G6PI) was added based on the gene *DNL43_RS04170* to enable proper glucose-6-phosphate interconversion.

These additions completed the core metabolic network essential for simulating heterofermentative metabolism.

We also identified incorrect reaction directions in the following key metabolic pathways: glycolysis, TCA cycle, nicotinamide metabolism, and amino acid biosynthesis. The reaction directions were corrected based on data from the KEGG, Model SEED, and BiGG databases, as well as the presence of the corresponding enzyme in the genome annotation of the *L. kefiri* DH5.

In particular, the directions of reactions involved in arginine and citrulline metabolism were corrected based on experimental data obtained for the closely related heterofermentative LAB, *Lactobacillus brevis* ATCC 367 [43], Notably, this metabolic route—the arginine deiminase (ADI) pathway—yields 1 mol of ATP per 1 mol of arginine consumed. Arginine is transported into the cell via an arginine/ornithine antiport, which facilitates the import of one arginine molecule in exchange for one ornithine molecule. The pathway consists of three consecutive enzymatic reactions: conversion of arginine to citrulline by arginine deiminase, transformation of citrulline into ornithine and carbamoyl phosphate via ornithine transcarbamylase, and eventually, the formation of ATP from ADP and carbamoyl phosphate in a reaction catalyzed by carbamate kinase [43]. However, the arginine/ornithine antiport was implemented in the base model with an incorrect direction for the transport reactions, which led to the formation of an arginine cycle and an excessive loss of ornithine relative to arginine uptake.

Following the implemented modifications, the growth rate predicted by the updated model was equal to 0, which was attributed to the excessive composition of the biomass equation and the inability to synthesize some of its components. Based on a comparison of the biomass composition of our model with published LAB models featuring experimentally validated biomass formulations [13,15,35], metabolites not present in other models were removed from the biomass equation. In addition, metabolites required for macromolecule synthesis (DNA, RNA, and protein) were decoupled into separate reactions. It should be noted that, due to the lack of experimental data on energy costs for their synthesis, ATP consumption was not included in these individual reactions at this step; instead, all ATP requirements, including those for macromolecule synthesis, remained consolidated within the biomass equation.

The modified model predicted a growth rate of 0.03 h^−1^, along with lactate and ethanol production (0.8783 and 0.828 mmol·gDCW^−1^·h^−1^, respectively). However, glucose uptake rate decreased to 1.317 mmol·gDCW^−1^·h^−1^, which contributed to the reduced growth rate. We hypothesized that this behavior is associated with an imbalance between NAD and NADP because the NAD to NADP ratio was approximately 2:1 in the modified model (total fluxes of 2.564 and 1.3 mmol·gDCW^−1^·h^−1^, respectively). To confirm this hypothesis, we introduced a pseudo-reaction NADTRHD (NAD transhydrogenase) into the model to maintain the balance between NAD and NADP. Its addition resulted in the NAD to NADP ratio shifted to 3:1 (total NAD and NADP fluxes of 35.271 and 12.473 mmol·gDCW^−1^·h^−1^, respectively). Furthermore, the lactate production rate became higher than the rate for ethanol in this case, which corresponds with the published experimental data [42]. At the same time, there is no enzyme which can catalyze these reactions in *L. kefiri*, and we attempted to identify an alternative mechanism in the strain metabolism which would lead to correct balance of NAD and NADP. An analysis of the *i*LM.c559 metabolic model showed that NAD/NADP balancing can be achieved by adding the NAD-dependent variant of glucose-6-phosphate dehydrogenase reaction, which is the main way of glucose oxygenation through the pentose phosphate pathway (PPP). However, according to published experimental data, this reaction mostly prefers NADP in Gram-positive bacteria, particularly in *L. mesentroides* [44]. Thus, we hypothesized the existence of an alternative glucose degradation pathway with NADH regeneration via D-gluconate (Figure 3). The model already contained NAD-dependent glucose 1-dehydrogenase (EC:1.1.1.47) and gluconate kinase reactions (EC:2.7.1.12). However, it lacked a reaction connecting D-glucono-1,5-lactone to D-gluconate (EC:3.1.1.17). Therefore, we added the gluconolactonase reaction (GL15LH reaction) to enable this pathway. It is also important to note that the model was unconstrained in the choice of classical and proposed shunts.

As a result, the alternative glucose degradation pathway we introduced became the primary route, with a flux of 23.184 mmol·gDCW^−1^·h^−1^, which corresponds to 93% of the glucose consumed by the bacterial cell according to the pFBA. In addition, the final model, compared to the base one, exhibited the production of both lactate and ethanol, with lactate being produced in higher amounts than ethanol (Figure 4), which is consistent with experimental data [42].

Additionally, to find out any experimental verification of our hypothesis about the functional activity of the proposed shunt, we re-analyzed the only published transcriptomics data [36] in which *L. kefiri* JCM5818 was co-cultivated with another LAB—*Lactobacillus kefiranofaciens* (details: Section 2.3). Analysis of the resulting heatmap of gene expression (Appendix A) at 37 °C, the optimal temperature for *L. kefiri* growth, shows the high level of expression of glucose 1-dehydrogenase (top five highly expressed genes). To prove that all reads mapped on the gene sequence can be assigned to *L. kefiri* JCM5818, we also checked for the presence of this enzyme in *L. kefiranofaciens* using NCBI BLASTp. The alignment analysis results showed that there are no enzymes highly expressed in the *L. kefiranofaciens* genome. Additionally, we analyzed the published meta-proteomics data from the supplementary material of the same study [36]. According to the proteomic data, this enzyme is present and functionally expressed, and annotated as an *L. kefiri* enzyme, which can, in total, indirectly confirm our hypothesis about the activity of the alternative NADH shunt in the pentose phosphate pathway in heterofermentative LAB. However, an enzyme activity assay is still required to shed light on the enzymatic activity of the reaction in the strain metabolism. Also, in the same study, the intracellular metabolomics data indicates the presence of gluconic acid (D-gluconate), for which synthesis enzyme 3.1.1.17 is missing, and its significant change between 30 and 37 °C (at 2 folds), which can indicate that this metabolite is produced in the cell [36]. According to the comparison of metabolic flux distributions between models with and without the NADH-dependent shunt, there are several metabolic differences which can be observed on metabolic maps constructed by the Escher tool (Appendix A). Firstly, shunt addition led to an activation of classical glucose transporter through proton symport in the final model, while only PEP-dependent transport reaction was active in the initial one. Moreover, a pyruvate kinase reaction (PYK) became active in the updated configuration of the metabolism. Expression of the enzyme was also confirmed by transcriptomics data (Appendix A). Eventually, activity of the Apartate, Asparagine, and lysine metabolic pathway is decreased in the model with the NADH-dependent shunt. The pathway had a high activity level in the base model as a compensatory mechanism for replenishment of the NADP pool in the cell, but most of the consumed aspartate (99%) was transferred from the cell as lysine, which could lead to an incorrect flux distribution in the model.

### 3.3. Selection of the Optimal Biomass Equation and Growth on Different Substrates

A number of genome-scale metabolic models, which were experimentally validated, have been reconstructed for some LAB strains actively used in biotechnology (Table 2). A comparative analysis of the *i*EM644 with other LAB models demonstrates that the proposed model has one of the largest number of genes, and the largest number of reactions and metabolites, which can be important for understanding the differences in LAB metabolism.

**Table 2 metabolites-15-00767-t002:** Comparison of published GSM models with experimental validation for LAB with *i*EM644.

Organism	Model ID	Genes	Reactions	Metabolites	Link
*L. casei* ATCC334 12A	*i*Lca334_548	548	1040	959	[45]
*L. casei* 12A	*i*Lca12A_640	640	1076	979	
*L. plantarum* WCFS1	*i*BT721	721	762	658	[13]
*L. reuteri* JCM 1112	Lreuteri_530	530	710	658	[35]
*L. mesenteroides* ATCC 8293	*i*LME620	620	762	754	[41]
*L. mesenteroides* subsp. *cremoris*	*i*LM.c559	559	1088	1129	[15]
*L. lactis* MG1363	*i*NF517	516	754	650	[14]
*L. kefiri* DH5	*i*EM644	644	1046	990	Present work

To identify the most influential components of the biomass composition, we performed a sensitivity analysis on the modified model via sequential alteration of the original stoichiometric coefficients of metabolites in the biomass reaction by multiplying them by 0.5 and 1.5 and evaluation of the impact of these changes on the predicted growth rate. Since the ATP requirement in the biomass equation is coupled to the production of ADP and phosphate, we treated the ATP hydrolysis (ATP + H_2_O → ADP + Pᵢ) as a single unit. Thus, its stoichiometric coefficients were scaled collectively during the sensitivity analysis. Whenever the coefficient of one of these metabolites was changed, proportional adjustments were applied to the entire group to preserve stoichiometric and energetic balance. The same approach was applied to peptidoglycan and teichoic acid components, as their formation is linked by interdependent reactions. Based on this analysis, we conclude that the model is most sensitive to changes in the coefficients of ATP and protein (changes more than 10% after increasing the coefficient) which is also compared to the results obtained by Kristjansdottir et al. [35] (Appendix A).

To further evaluate the impact of alternative biomass formulations, we selected three experimentally validated biomass equations from published GSM models of heterofermentative *Lactobacillaceae* family members (Table 2): *L. plantarum* WCFS1 (*i*BT721), *L. reuteri* JCM 1112 (Lreuteri_530), and *L. mesenteroides* subsp. *cremoris* (*i*LM.c559). We subsequently assessed the effects of substituting biomass components using formulations from *L. mesenteroides* subsp. *cremoris*, *L. plantarum* WCFS1, and *L. reuteri* JCM 1112. We specifically replaced the coefficients for ATP, protein, DNA, and RNA with values taken from these models. Stoichiometry for peptidoglycan and teichoic acids were left unchanged, as their precise composition requires experimental verification and cannot be directly transferred from other models. In parallel, we modified the DNA, RNA, and protein biosynthesis reactions to match those in the *i*LM.c559, *i*BT721, and Lreuteri_530 models, preserving metabolite composition and stoichiometric coefficients. Although changes in DNA and RNA coefficients did not significantly affect the growth rate (due to the absence of energy costs for their synthesis in our model), we integrated the corresponding biosynthesis reactions, including their energy requirements.

All simulations were performed under standardized conditions: uptake rates of extracellular metabolites were constrained according to the CDM formulation, with glucose uptake fixed at 25 mmol·gDCW^−1^·h^−1^. Results of the parsimonious flux balance analysis (pFBA) are presented in Figure 5 and Appendix A.

Analysis of flux values (Figure 5A) revealed that model variants exhibited variation in growth rates depending on the source of the biomass equation. The model incorporating the biomass equation from *L. reuteri* JCM 1112 demonstrated the highest growth rate, while models using biomass equations from *L. mesenteroides* subsp. *cremoris* and *L. plantarum* WCFS1 showed similar growth rates. It is important to note the close phylogenetic relationship among *L. kefiri*, *L. reuteri*, *L. mesenteroides*, and *L. plantarum* [46]. At the same time, *L. kefiri*, *L. reuteri*, and *L. mesenteroides* are obligately heterofermentative [47], whereas *L. plantarum* is facultatively heterofermentative [48]. Therefore, we suggested that the use of the *L. plantarum* biomass equation may be biologically unjustified due to metabolic differences.

*L. kefiri* exhibits biphasic growth, as reported by Kondybayev and co-authors [42]: the growth rate reaches 0.20 h^−1^ in the first phase and increases to 0.35 h^−1^ in the second phase. It should be emphasized that the transition between phases depends on cultivation temperature and medium pH. According to the data presented in Figure 5A, the *L. kefiri* model using the biomass equation from *L. mesenteroides* most closely matches the experimentally observed growth rate during the second phase. It should be emphasized that although experimental data are available, the strain of *L. kefiri* used in the study was not specified. Moreover, the cultivation in the study was carried out in MRS medium, which slightly differs from CDM content, but it highlights the necessity for additional experiments using the *L. kefiri* DH5 strain growing in a defined CDM to measure carbon source uptake rates and the corresponding growth rate.

Differences were also observed in flux rates of secreted metabolites: models with biomass equations derived from Lreuteri_530 and *i*LM.c559 demonstrated complete glucose consumption and nearly identical production rates of carbon dioxide, lactate, and ethanol (Figure 5B). It should also be noted that the automatically Kbase-generated biomass equation yielded the similar result (Figure 5B). Meanwhile, despite comparable growth rates between the model with the *i*BT721-derived biomass equation and the model with the *i*LM.c559-derived biomass equation, the former exhibited reduced glucose uptake and lower production rates of CO_2_, lactate, and ethanol (Figure 5B).

We performed a sensitivity analysis on the biomass equation for all three model versions, following the methodology outlined in Section 2.5. The models using different biomass equations exhibited distinct sensitivity patterns to variations in the content of key components, including ATP, peptidoglycan, teichoic acids, protein, DNA, RNA, and water (Appendix A). The highest sensitivity was observed in the model with the biomass equation derived from Lreuteri_530 regarding protein content. A decrease in the corresponding stoichiometric coefficient led to a 32.1% increase in the growth rate, while an increase resulted in a 19.6% decrease relative to the baseline.

The *i*BT721 model also showed substantial growth rate changes in response to variations in the protein coefficient: a decrease increased the growth rate by 20.4%, and an increase decreased it by 22.4%. Furthermore, all models responded to changes in the coefficient for ATP and its hydrolysis products. A decrease in the coefficient increased the growth rate by 10.3% (Lreuteri_530) and 11.8% (*i*LM.c559), whereas an increase reduced it by 8.6% (Lreuteri_530), 11.0% (*i*BT721), and 18.4% (*i*LM.c559).

Notably, the models based on Lreuteri_530 and *i*LM.c559 exhibited a measurable influence of the RNA coefficient on the growth rate. For Lreuteri_530, decreasing the coefficient increased the growth rate by 3.3%, and increasing its value decreased the growth rate by 3.3%. The corresponding changes led to +1.9% and −1.8% changes in the growth rate for *i*LM.c559. The impact of the DNA coefficient was negligible.

In contrast, the *i*BT721 model displayed a pronounced sensitivity to the content of peptidoglycan and teichoic acids, which was not shared by the other two models. A decrease in the coefficient increased the growth rate by 11.9%, and an increase reduced it by 12.4%. The sensitivity to these components in the other two models was approximately four times lower.

A sensitivity analysis was also performed for the ATPM reaction, which represents the non-growth-associated maintenance for each model version. Due to the lack of experimental NGAM data for *L. kefiri* DH5, we assessed the impact of different lower bounds for the ATPM flux on the growth rates of models with different biomass equations. The lower bounds were set to 0, 0.36, and 0.51 mmol·gDW^−1^·h^−1^, corresponding to the experimentally determined NGAM values for the *i*BT721 (0.36 mmol·gDW^−1^·h^−1^) and *i*LM.c559 (0.51 mmol·gDW^−1^·h^−1^) models. The analysis revealed that the model utilizing the *i*BT721 biomass equation did not alter its growth rate in response to changes in the NGAM constraint. In contrast, the models with the *i*LM.c559 and Lreuteri_530 biomass equations exhibited a minor decrease in growth rate: a 1.9% reduction at an NGAM value of 0.36 and a 2.6–2.7% reduction at a value of 0.51 (Appendix A).

Therefore, we selected the *L. kefiri* model incorporating the biomass equation from *i*LM.c559, with the lower bound for the ATPM reaction set to 0.51 mmol·gDW^−1^·h^−1^, as in the original *i*LM.c559 model, for all subsequent analyses.

### 3.4. Metabolite Exchange Profile of the L. kefiri DH5 Model Under Different Substrate and Aerobic Conditions

Metabolite exchange profiles obtained using FVA for growth on glucose as a carbon substrate showed comparable and wide ranges of production of key metabolites of the heterofermentative pathway: lactate, ethanol, and CO_2_ [15]. To assess the metabolic flexibility of the model, we used flux variability analysis (FVA) to determine the range of feasible fluxes for key excreted products of heterofermentative metabolism (lactate, ethanol and CO_2_) during growth on glucose. The resulting flux ranges were wide and comparable, indicating significant metabolic plasticity. This profile agrees with that predicted by the model *i*LM.c559 [15]. However, the flux ranges for these same metabolites in *i*LM.c559 were narrower (Appendix A). The observed ratio of fermentation products—lactate/ethanol/CO_2_—is comparable to the experimental data obtained for *L. mesenteroides* and amounted to 1:1:1 [15]. At the same time, a slight shift towards greater production of lactate compared to ethanol was noted, which also agrees with the experimental data. In contrast to glucose and lactose, the model did not demonstrate significant growth on galactose (growth rate ~0.002 h^−1^). Moreover, the strict limitation of flux boundaries on galactose uptake rate led to an infeasible solution.

The metabolic flux distribution under aerobic conditions was also assessed by the model simulation. To correctly model the aerobic fermentation, the direction of the ATPase reaction was set towards the generation of ATP (lower and upper bounds—[−1000, 1000]). It is known that the acetate produced in some lactic acid bacteria under anaerobic conditions is utilized for acetyl-phosphate synthesis, which feeds into ethanol formation to maintain cellular redox balance [15]. Furthermore, some research demonstrates that a shift to aerobic conditions redirects metabolism from ethanol production towards acetate secretion. This mechanism is linked to the aerobic oxidation of NADH [49,50]. We observed the phenomenon via the model simulations (Figure 6): an increase in oxygen uptake led to a metabolic shift from ethanol to acetate production, which is consistent with the result described in the article by Ozcan and co-authors [15]. It is important to note that an increase in the oxygen uptake rate did not have a significant effect on the production of CO_2_ and lactate (Figure 6). The model analysis outcome also completely agrees with the results described by Ozcan and co-authors [15], where no correlation between their production rates and oxygen consumption was observed (Appendix A). We also made a 3D phase-space plot in which the feasible flux ranges for acetate, ethanol, and acetyl-phosphate are shown as oxygen-dependent trajectories (Appendix A). FVA results show that the minimal acetyl-phosphate flux decreases in parallel with a decrease in ethanol production as oxygen availability increases. At the same time, despite the maximal ethanol flux also declining, the maximal acetyl-phosphate flux increases, indicating greater intracellular availability of this metabolite in these conditions, corresponding to the earlier notion that acetyl-phosphate and ethanol production are metabolically linked.

## 4. Discussion

### 4.1. Tool-Driven Specificity of Generated Genomes-Scale Metabolic Models for L. kefiri: From Comparative Analysis to Cons and Pros

Although there are studies focused on the comparative analysis of different GSM reconstruction tools considering their features, these overviews cover only a limited number of available computational methods. Among tools used in the present study, only KBase and CarveMe have been previously evaluated [24,51,52], whereas no such comparisons have been conducted for PanGEM, Bactabolize, and Reconstructor. We identified significant differences between metabolic models for *L. kefiri* DH5 built by different tools. These distinctions boil down to both the size of the models (the total number of reactions, metabolites, and genes) and the composition of the reactions added during the gap-filling process. These characteristics were directly reflected in the MEMOTE score for each model. The model reconstructed with the Reconstructor tool included a greater number of genes compared to other tools but at the same time contained one of the lowest numbers of reactions. This is likely related to the specific features of the gap-filling algorithm employed in Reconstructor, which is based on pFBA and assumes minimization of the total metabolic flux. Despite the theoretical advantage of this approach, it did not take into account phylogenetic relatedness and enzyme homology, which can lead to biologically incorrect additions. In particular, in a number of cases, we observed the addition of reactions from eukaryotic organisms, which is methodologically unfounded for reconstructing the metabolism of a prokaryotic cell. As a result, the model built using Reconstructor contained only 402 genes corresponding to *L. kefiri* DH5 (Figure 1), which was one of the lowest values among all considered models. Metabolic models reconstructed by the KBase and CarveMe tools demonstrated the best quality among all model versions for *L. kefiri* DH5. It is worth noting that the model developed by KBase showed higher-quality indicators, especially in terms of mass and charge balance, than the CarveMe-driven model. Both models contained a comparable number of genes but differed significantly in the number of reactions: the model obtained using CarveMe included 80% more reactions than the KBase model. A significant difference was also observed in the fraction of reactions added during gap-filling: the CarveMe-generated model accounted for 28% of such reactions from the total, while the KBase model accounted for only 12%. This indicates differences in gap-filling strategies between the tools, which can lead to non-specific model behavior and reduce predictive power. At the same time, tools based on the pangenome approach demonstrated the worst reconstruction quality. The low MEMOTE scores were primarily due to problems with stoichiometric consistency and lack of mass balance for 26% of reactions in the PanGEM model and 25% in the Bactabolize-generated version, as well as lack of charge balance in 6% and 8% of reactions represented in the PanGEM and Bactabolize versions, respectively. In addition, these models showed the worst values for the unbounded flux in default medium criterion: 95% of reactions were without bounds in PanGEM and 77% were in Bactabolize, indicating potential problems in defining the feasible solution space within FBA. It is also noteworthy that both models reconstructed by the pangenome-based approach contained the largest number of metabolites—1695 in each model. This may be due to the fact that tools using the pangenome approach aim to consider the metabolic features of the entire genus rather than of a specific organism. This leads to the inclusion of a large number of compounds potentially not characteristic of the specific bacterium but present in related species, which ultimately limits the applicability of such models for strain-specific metabolic predictions.

It should also be noted that the models reconstructed using KBase, CarveMe, and PanGEM contained the highest number of unique genes: the KBase model included 115 such genes; CarveMe, 129; and PanGEM, 78. Matching these genes to their corresponding metabolic pathways revealed characteristic features of each model. Despite the CarveMe model having the largest total number of unique genes, most of them were annotated as encoding transporters, including proteins involved in magnesium and zinc transport. Genes related to folate biosynthesis were also identified in the list of CarveMe unique genes (Appendix A). The PanGEM model contained fewer unique genes, while they covered a broader range of metabolic pathways: transporter-encoding genes predominated among them, but genes involved in the metabolism of riboflavin, flavin mononucleotide, and flavin adenine dinucleotide, as well as in threonine and homoserine biosynthesis and the Entner–Doudoroff pathway, were also present (Appendix A). The model reconstructed with KBase demonstrated the greatest functional completeness: unique genes were predominantly related to the biosynthesis of macromolecules, folate, and histidine, as well as to the utilization of glycine, serine, proline, and 4-hydroxyproline (Appendix A). This indicates a more comprehensive coverage of metabolic capabilities in the KBase model compared to the other tools. Nevertheless, the unique genes from the CarveMe and PanGEM models may be useful for further refinement of the KBase model.

### 4.2. NADH and NADPH Recovery in Heterofermentative Bacteria

The imbalance between NAD and NADP that was identified in the model reconstructed using KBase became an important point that prompted us to search for an alternative way for glucose metabolism in the pentose phosphate pathway. Restoring this balance by introducing a pseudo-reaction of transhydrogenase formally resolved the issue but lacked biological justification, as no enzyme catalyzing such a reaction has been identified in *L. kefiri*. An imbalance in redox cofactors within the reconstructed model can lead to unrealistic thermodynamic cycles, such as non-physiological NADP regeneration. Similar issues were observed during the reconstruction of the GSM model for *Leuconostoc mesenteroides*, where the authors reported the same challenge [15]. To address this, Özcan et al. introduced an NAD-dependent variant of glucose-6-phosphate dehydrogenase (G6PDH) based on studies indicating that the enzyme can form a complex with NAD despite its preference for NADP [44,53,54]. This NAD-dependent variant follows a steady-state random mechanism, where the enzyme can bind NAD, NADP, and glucose-6-phosphate (G6P). However, if the enzyme binds G6P first, it becomes specific to NAD and forms a dead-end complex for NADP-dependent activity [53]. Nevertheless, kinetic studies on this enzyme are lacking for *Lentilactobacillus* spp. and other heterofermentative LAB. In contrast, the enzyme in *Lactobacillus casei*, a homofermentative LAB, is confirmed to be NADP-specific [55]. We also analyzed the effect of NAD(P) transhydrogenase, which can transfer protons between NAD and NADP. There is no gene for this enzyme in *L. kefiri,* but members of the genus *Lentilactobacillus* possess the *pntA* and *pntB* genes encoding the NAD(P) transhydrogenase (EC 1.6.1.1). Interestingly, most species of the genus harbor only *pntB*, which encodes the β-subunit of the enzyme, whereas closely related species such as *Lentilactobacillus sunkii*, *Lentilactobacillus buchneri*, and *Lentilactobacillus parakefiri* contain both *pntA* and *pntB*. The presence of this enzyme can be associated with high ethanol tolerance [56]. Specifically, *pntA* encodes the NAD-dependent subunit, and *pntB* the NADP-dependent one. The co-occurrence of both enables proton translocation across the membrane. However, this enzyme is not typical for the *Lactobacillaceae* family as a whole. The observed differences in metabolic organization between heterofermentative and homofermentative LAB are known to be linked with lineage-specific gene loss or gain events in the pentose phosphate and Embden–Meyerhof–Parnas pathways [57]. The occurrence of *pntAB* in some *Lentilactobacillus* strains could be an adaptive mechanism under ethanol-rich environmental conditions. Furthermore, during LAB evolutionary adaptation, several alternative mechanisms might have evolved to maintain the NAD/NADP balance.

Due to the uncertainty around these mechanisms, an alternative metabolic route was explored, leading to the identification of a shunt involving NAD-dependent glucose 1-dehydrogenase (reaction rxn01108_c0), which supports additional NADH regeneration. A limitation of this pathway was that the product of this reaction, D-glucono-1,5-lactone, could not be metabolized further in the original model due to the absence of its hydrolysis to D-gluconate. Notably, the subsequent reaction—gluconate phosphorylation (GNKr)—was already present in the model according to the genome annotation. To resolve this gap, a candidate gene (*DNL43_RS07155*), annotated as a lactonase, was identified in the *L. kefiri* DH5 genome and is likely capable of catalyzing this hydrolysis step. This reaction can also be catalyzed by other hydrolases, which improves the robustness of the pathway under diverse conditions. Furthermore, the enzyme’s promiscuity is described as an effective adaptive strategy in bacteria [58,59], and such flexibility may support the role of *DNL43_RS07155*, annotated as 6-phosphogluconolactonase (EC:3.1.1.31), in catalyzing this reaction. This assumption relies on substrate promiscuity, whereby an enzyme can act on different substrates through the same catalytic mechanism [60,61,62,63,64]. Catalytic promiscuity may also contribute to this reaction, as enzymes can catalyze transformations beyond their primary annotation [60,61,62,63,64], although the possibility of such catalytic activity toward D-glucono-1,5-lactone hydrolysis requires further confirmation. In addition, spontaneous hydrolysis of D-glucono-1,5-lactone has been reported in some Gram-negative bacteria, indicating that this conversion does not necessarily require enzymatic catalysis [65,66,67].

The role of glucose 1-dehydrogenase also gained further support from transcriptomic and proteomic data (see Section 3.1), showing high expression of the corresponding gene (*DNL43_RS11740*) in *L. kefiri*. This suggests that the proposed alternative glucose degradation pathway is likely active in vivo. The shunt provides additional NADH that gives an opportunity to in silico reproduce key metabolic outputs characteristic of heterofermentative LAB: the production of lactate in higher quantities than ethanol according to the experimental data [42]. Importantly, this enzyme is known to utilize both NAD and NADP and plays a role in redox balancing [68,69,70,71]. However, no study to date has explored the function of this enzyme in LAB, and this alternative pathway has not been considered in either automated reconstructions of *L. kefiri* models or published models of other LAB. BLASTp analysis shows that this gene is present in various LAB species, including *Lentilactobacillus raoultii*, *Secundilactobacillus oryzae*, *Pediococcus acidilactici*, and *Lentilactobacillus buchneri*, for which GSM models have not yet been reconstructed. Moreover this gene is located on a plasmid according to the *L. kefiri* genome annotation, suggesting it may not be universally present across all heterofermentative LAB.

Taken together, these findings support the hypothesis that some LAB possessing this enzyme (EC:1.1.1.47) may utilize a more robust NADH-regeneration pathway via glucose 1-dehydrogenase, as opposed to the less stable NAD-dependent G6PDH mechanism (EC:1.1.1.49). This pathway could represent an overlooked aspect of LAB metabolism with potential implications for improving their biotechnological traits. The computational results require further experimental validation to confirm their functionality and significance.

### 4.3. Impact of the Biomass Equation Stoichiometry on the Model Prediction Results

One of the key factors affecting simulation results is the choice of the biomass equation’s source. Several studies [72,73,74] have shown that model behavior largely depends on the composition of the biomass equation and the stoichiometry coefficients used. At the same time, there is a limited amount of experimental data on biomass composition for LAB. It should be noted that biomass composition can vary significantly depending on the strain. However, the biomass equation is not frequently constructed based on experimental data obtained specifically for a given organism [73]. The automatically generated biomass equation for our base model of *L. kefiri* DH5, which was reconstructed by KBase, was a generalized biomass equation for Gram-positive bacteria [75]. This approach can significantly distort the model predictions.

To evaluate the impact of biomass composition and associated energy costs on model predictions, we adopted experimentally validated biomass equations and macromolecule synthesis reactions from models of heterofermentative representatives of the microbial family *L. reuteri* JCM 1112, *L. plantarum* WCFS1, and *L. mesenteroides* subsp. *cremoris*. It should be noted that, although all these organisms are phylogenetically related to *L. kefiri*, only *L. reuteri* JCM 1112 and *L. mesenteroides* subsp. *cremoris* are obligate heterofermenters, making their biomass formulations more metabolically appropriate for *L. kefiri* DH5.

Simulation results indicated that the choice of biomass equation had a stronger influence on predicted growth rates than on the exchange profiles of key LAB metabolites. In all cases, lactate production exceeded ethanol production, consistent with experimental observations. Nevertheless, we selected the biomass equation from the *L. mesenteroides* subsp. *cremoris* model as the most suitable for *L. kefiri* DH5 based on the comparative analysis. This was primarily caused that both *L. reuteri* JCM 1112 and *L. mesenteroides* subsp. *cremoris* are obligate heterofermenters, lacking two key glycolytic enzymes—phosphofructokinase and fructose-1,6-bisphosphate aldolase—and metabolize glucose via the phosphoketolase pathway. However, the model variant incorporating the *L. reuteri* JCM 1112 biomass equation exhibited the highest growth rate among all tested variants (0.47 h^−1^), which does not align with experimental data for *L. kefiri* DH5 (0.35 h^−1^) [42]. Therefore, we prioritized the *L. mesenteroides* subsp. *cremoris* formulation. The *L. plantarum* WCFS1-derived biomass equation was excluded from further consideration due to the facultative heterofermentative metabolism characteristic of *L. plantarum*.

In addition to ATP costs associated with growth and macromolecule synthesis, non-growth-associated maintenance (NGAM) also contributes to cellular energy demand. However, our sensitivity analysis of NGAM across the three model variants revealed minimal impacts on predicted growth rates. Consequently, we adopted an NGAM lower bound of 0.51—a value taken from the *L. mesenteroides* subsp. *cremoris* model and experimentally validated. Under this configuration, the final model predicted a growth rate of 0.36 h^−1^, which was in close agreement with experimental measurements for *L. kefiri* DH5.

Adjustment of the biomass equation also resulted in the reproduction of the typical metabolic exchange profile for heterofermentative LAB where lactate production exceeds that of ethanol. These observations highlight the necessity of experimental verification of biomass composition: both at the level of macromolecular composition (proteins, RNA, DNA, and lipids) and at the level of elemental and energetic balances. The data would not only improve the accuracy of the model but also reduce uncertainty in reconstructing pathways involved in redox balance and cellular energy metabolism.

### 4.4. Impact of Galactose and Lactose Metabolism on Predicted Growth

The model behavior during growth on galactose and lactose exhibited notable differences. The metabolic route of galactose—specifically, its conversion to glucose-6-phosphate—bypasses the alternative NADH-generating shunt proposed in our model. This results in a disruption of redox balance, leading to reduced growth rate, substrate uptake, and metabolite production. Nevertheless, even with both low growth and galactose uptake rates, the exchange profiles of lactate, ethanol, acetate, and CO_2_ remain specific and correspond to the observed one for heterofermentative LAB. Since lactose metabolism involves galactose liberation, it similarly results in reduced growth rate. To test the hypothesis on redox imbalance, we introduced the NAD transhydrogenase reaction (NADTRHD) into the model, setting its flux bounds to [–1000, 1000]. During growth on lactose, no significant changes in metabolic exchange profiles were observed, while the value of growth rate was doubled (Appendix A). More pronounced differences were observed during growth on galactose: without active NADTRHD, the model predicted a low level of the galactose consumption rate. The metabolic exchange profile plateaued at low yields of key products under its maximum uptake (1.207 mmol·gDW^−1^·h^−1^). When the NADTRHD route was activated, the model demonstrated higher growth rate, increased galactose uptake, and enhanced metabolite production while preserving heterofermentative exchange profiles (Appendix A). This behavior underscores the necessity of further experimental validation of growth rates, substrate consumption, and metabolic exchange profiles to identify physiologically grounded mechanisms for restoring intracellular redox balance.

## 5. Conclusions

We present the first curated genome-scale metabolic model of *Lentilactobacillus kefiri* DH5 and a systematic comparison of current reconstruction pipelines for heterofermentative LAB. Our analysis shows that tool selection strongly affects model quality and predictive power.

Through extensive refinement, we identified a redox imbalance that prevented realistic simulation of heterofermentative metabolism. Incorporation of an alternative NADH-regenerating glucose shunt via D-gluconate restored physiological fluxes and is supported by omics data, suggesting a previously overlooked redox-balancing mechanism in *L. kefiri*.

We further demonstrated that biomass composition significantly shapes growth predictions, with the *Leuconostoc mesenteroides*-derived biomass equation yielding the most biologically plausible behavior. The final model, *i*EM644, robustly captures substrate-dependent and oxygen-driven metabolic shifts characteristic of heterofermentative LAB.

Overall, this work provides a high-quality metabolic framework for *L. kefiri* and highlights the importance of integrated curation, redox balancing analysis, and informed biomass selection for accurate GSM reconstruction.

## Figures and Tables

**Figure 1 metabolites-15-00767-f001:**
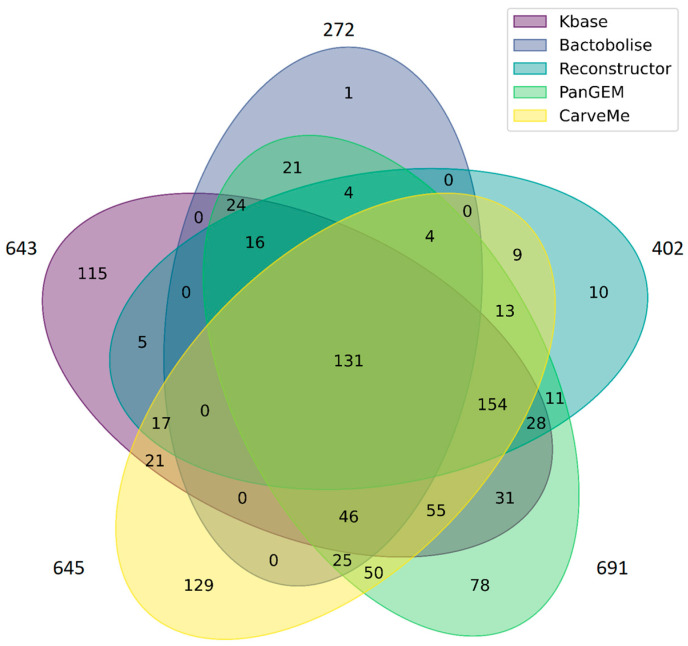
Venn diagram shows the gene overlap between models reconstructed by different tools.

**Figure 2 metabolites-15-00767-f002:**
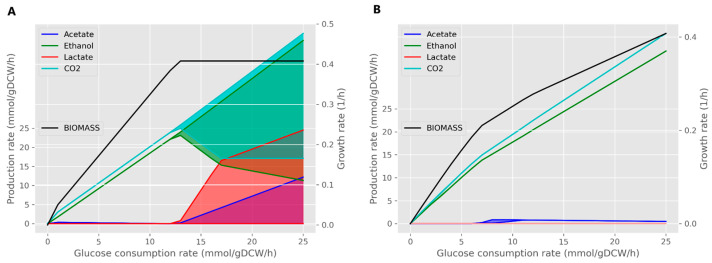
Flux variability analysis (FVA) of reconstructed models on CDM with glucose as a main carbon source. (**A**) FVA of KBase model reconstruction; (**B**) FVA of CarveMe model reconstruction.

**Figure 3 metabolites-15-00767-f003:**
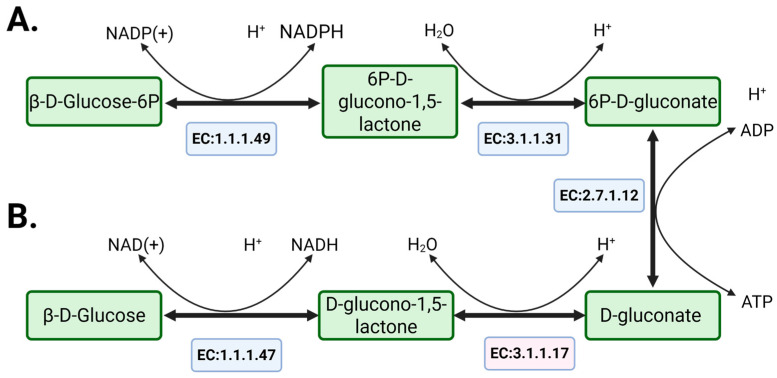
Reactions in the pentose phosphate pathway associated with 6-phospho-D-gluconoate synthesis. (**A**) Classical reactions cascade; (**B**) hypothesis of potential shunt through D-gluconate. Blue rectangles highlight enzymes which are present in the *L. kefiri* genome, and red highlights those which are missing. Created in (https://BioRender.com).

**Figure 4 metabolites-15-00767-f004:**
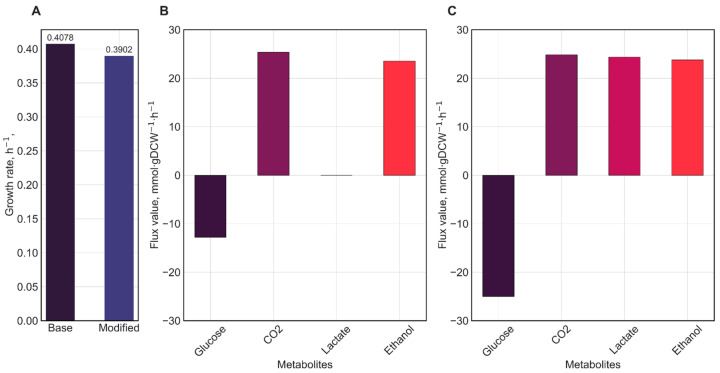
Comparison of growth rates, glucose consumption, and metabolite production between the base model and the final modified model. (**A**) Comparison of growth rates between the base and modified models. (**B**) Glucose consumption and production of key LAB metabolites in the base model. (**C**) Glucose consumption and production of key LAB metabolites in the modified model.

**Figure 5 metabolites-15-00767-f005:**
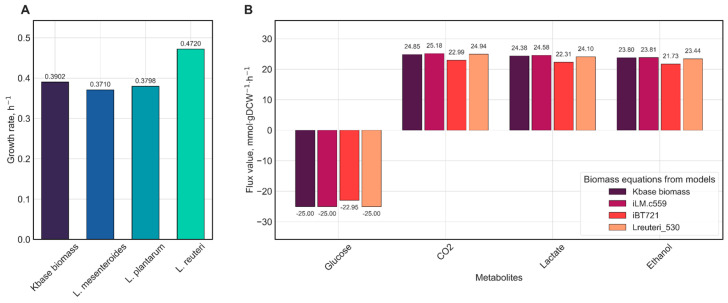
Comparison of simulation results for *i*EM644 model versions with biomass equations from *L. mesenteroides* subsp. *cremoris* (*i*LM.c559), *L. plantarum* WCFS1 (*i*BT721) and *L. reuteri* JCM 1112 (Lreuteri_530). (**A**) Comparison of growth rates in the *i*EM644 model using different biomass equations. (**B**) Comparison of glucose consumption and production of key LAB metabolites in the *i*EM644 model using different biomass equations.

**Figure 6 metabolites-15-00767-f006:**
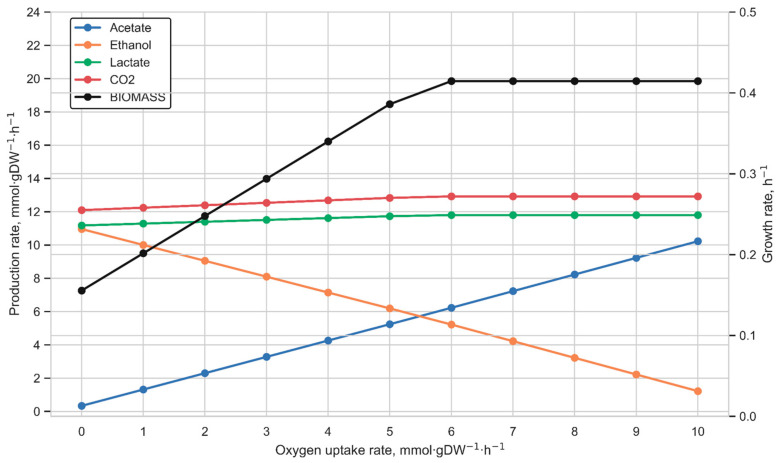
Model predictions of the metabolic shift between ethanol and acetate production by increasing the oxygen uptake rate during growth in CDM and at a fixed glucose uptake rate of 10 mmol·gDW^−1^·h^−1^.

**Table 1 metabolites-15-00767-t001:** Statistics of reconstructed models for *L. kefiri* DH5 using different automated reconstruction tools.

Model	Genes	Reactions (Reactions Without GPR)	Metabolites	MEMOTE, %	
				Total Score	Sub Total
Bactabolize	372	798 (11)	1695	20	35
**CarveMe**	655	1871 (532)	1279	**87**	93
**KBase**	643	1037 (124)	994	84	**95**
PanGEM	691	1313 (329)	1695	29	36
Reconstructor	741	1006 (110)	1095	83	93

The bold highlights indicate the models that demonstrated the best MEMOTE score.

## Data Availability

Appendix A and models are available on GitLab via the link https://gitlab.sirius-web.org/biotech_lactobacteria/l_kefiri_dh5_model (accessed on 17 November 2025), as well as on BioUML via the link https://uni.sirius-web.org:58443/bioumlweb/#de=data/Collaboration/FT_LAB_project/Data/Modeling/L_kefiri_DH5 (accessed on 17 November 2025).

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
