# Peer review of "Evaluation of Genome-Scale Model Reconstruction Strategies for Lentilactobacillus kefiri DH5 and Deciphering Its Metabolic Network"

_metabolites, 2025, doi:10.3390/metabo15120767_

Round 1

Reviewer 1 Report

Comments and Suggestions for Authors

Review for metabolites

The manuscript by authors reported a detailed construction of a metabolic model of a LAB Lentilactobacillus kefiri DH5. In general, the reconstruction procedures are well-characterized and the comparison of current (semi)automatic software are useful. The reconstruction details are displayed in the manuscript. I think this manuscript have higher standard in GSM construction and deserved publishing. Following are some suggestions for the authors to consider.

Major comments

  1. In many opinion, the information and background in Introduction is sufficient. However, this section are in one paragraph, which may be hard to follow. I suggested change it into two or three paragraph. In addition, add some sentence/information relevant to the why experimental-based researches are still not enough to understand the healthy impact could be useful.
  2. Result 3.4.  I am wondering whether the metabolite generation/excretion rates (fluxes) obtained by FVA can be described as ‘metabolic profile’since it may mislead the readers as the experimental measured concentration/content of metabolites, such as metabolome. Why the growth rate is low when galactose is the carbon source ? Add some analysis and discussion on this result could be useful, such as the theoretical growth rate according to the meatbolic routes of galactose. In kefiri, is the galactose are redirected into glycolysis or it lead to other metabolic pathway ?
  3. Figure 6 and related text: This topic is very interesting and have deepened the understanding on regulation on ethanol formation and acetate utilization. A heatmap described a phenotype phase plane analysis on oxygen absorb rate, acetate and ethanol production rates can be stressed this finding.
  4. Discussion 4.2  The authors put forward a interesting and important scientific question here - the reduction force imbalance in LAB. I listed some suggestions for authors to consider, which I think can enhanced the manuscript. First, the author focus on adding a pseudo-reaction that catalyzing the inter-conversion of NADP+ and NAD+, which endow the cell ability to regulate the NAD/NADP ratio. I agree the authors that the non-GPR state on this reaction in LAB GSMs are leave this enzymatic reaction in unconstrained states, leading to unrealistic prediction. However, I think here, further discussion can be added. What is the most close related strains (in phylogeny) having the genes involved in this reaction ? All Lactobacillus lacked this gene ? If so, what the possible biological meaning on this ?
  5. Discussion 4.4: The topic here is also interesting. However, why the authors discussed the metabolic behavior on galactose and lactose ? What is the biological meaning on galactose and lactose metabolism in LAB?

Minor comments and typos

  1. Line 48: ‘adhE’gene symbols should in italic.
  2. Line 586 ‘coathours’could be a wrong typo
  3. Line 602 ‘iBT721’the letter i should in italic.
  4. Line 655-656, the species name should in italic.
  5. Line 654-657: The authors stated the prediction on fermentation product are consistent of the experimental data. I think the reference should be added here. And here, when carried out the FVA, what the boundary used of other metabolites ? It could be useful to report the boundaries, especially the oxygen boundaries since it affected the metabolic flux allocation.

In sum, I think this work is well carried out and written. I enjoy reviewing this manuscript and looking forward to see the manuscript being published.

Reviewer 2 Report

Comments and Suggestions for Authors

  1. The study employedfive tools but provides limited rationale for selecting this specific set. A brief discussion on why these tools were chosen over others would be beneficial.
  2. The author proposed an alternativeNADH-regenerating glucose shunt for redox balance regulation. How about its effect on intracellular NADH/NAD+ level?
  3. The rationale for selecting the KBase model over CarveMe is based on its ability to produce a heterofermentative profile (lactate, ethanol, CO2). However, the reason why the CarveMe model failed to do so is not thoroughly discussed.
  4. The introduction could be more focused on the specific gaps in L. kefiri metabolism knowledge and how GEMs can address them, rather than providing a very general overview of GSM utility.
Comments on the Quality of English Language
  1. The manuscript would benefit from professional proofreading to correct minor grammatical errors and improve the overall fluency of the writing.

Reviewer 3 Report

Comments and Suggestions for Authors

This manuscript constructed different computational models for Lentilactobacillus kefiri DH5, where the iEM644 model enabled realistic lactate and ethanol production. I have the following concerns:

  1. It is still unclear to me how likely there would be a shunt pathway through D-gluconate since the required enzyme (EC:3.1.1.17) is missing. Please provide a better explanation or clear evidence to support it.
  2. If there is such a shunt pathway, can you change the flux of this pathway to different levels and predict how the NAD/NADP ratio influences lactate/ethanol production?

Round 2

Reviewer 3 Report

Comments and Suggestions for Authors

Thank you for addressing my concerns. I have no further comments.